# Insights into the Root Invasion by the Plant Pathogenic Bacterium *Ralstonia solanacearum*

**DOI:** 10.3390/plants9040516

**Published:** 2020-04-16

**Authors:** Hao Xue, Rosa Lozano-Durán, Alberto P. Macho

**Affiliations:** 1Shanghai Center for Plant Stress Biology, Center for Excellence in Molecular Plant Science, Chinese Academy of Sciences, Shanghai 201602, China; haoxue@psc.ac.cn; 2University of the Chinese Academy of Sciences, Beijing 100049, China

**Keywords:** *Ralstonia*, plant defence, bacterial colonization, root invasion, plant-bacteria interactions

## Abstract

The plant pathogenic bacterium *Ralstonia solanacearum*, causal agent of the devastating bacterial wilt disease, is a soil-borne microbe that infects host plants through their roots. The initial mutual recognition between host plants and bacteria and the ensuing invasion of root tissues by *R. solanacearum* are critical steps in the establishment of the infection, and can determine the outcome of the interaction between plant and pathogen. In this minireview, we will focus on the early stages of the bacterial invasion, offering an overview of the defence mechanisms deployed by the host plants, the manipulation exerted by the pathogen in order to promote virulence, and the alterations in root development concomitant to bacterial colonization.

## 1. Introduction

*Ralstonia solanacearum* is one of the top ten plant pathogenic bacteria worldwide according to its scientific and economic importance [1]. *R. solanacearum* is the causal agent of bacterial wilt disease in more than 250 plant species, including agriculturally important crops such as tomato, potato, banana, and peanut [1,2], and can also infect model plants, such as *Arabidopsis thaliana* (hereafter, Arabidopsis) and *Medicago truncatula* [1]. As a soil-borne pathogen, *R. solanacearum* enters plants through the root, using wounds, root tips, and secondary root emerging points as penetration sites; it then progresses via the root cortex, finally reaching the vascular system [3,4,5], as shown in Figure 1. From this point onwards, and mostly through xylem vessels, the infection spreads systemically in the plant [5,6]. The invading bacteria multiply massively in the xylem and produce abundant exopolysaccharides (EPSs), which ultimately leads to the obstruction of the vessels and the subsequent development of the typical wilting symptoms due to impaired water conductance [7].

*R. solanacearum* is a devastating pathogen with a dramatic economic impact worldwide. Gaining a deeper understanding of the molecular and physiological mechanisms underlying the pathogenicity of this bacterial species is a necessary stepping stone in the design of effective and durable strategies for crop protection in order to guarantee food security. Considering that *R. solanacearum* is present in the soil, the initial recognition between host plants and bacteria and the ensuing invasion of root tissues are crucial steps in the establishment of the disease, and as such deserve special attention. In this minireview, we will focus on the early stages of the bacterial infection, providing an overview of the defence mechanisms deployed by the host plants upon recognition of the bacteria, of the manipulation exerted by the pathogen in order to promote its own multiplication and spread, and of the developmental changes occurring in the root system during bacterial colonization.

## 2. Plant Defence Responses Encountered by *R. solanacearum*

Plants are constantly exposed to potential pathogens and therefore, in order to protect themselves, they have evolved a complex and multilayered immune system, which ultimately succeeds in defending a given plant species from most invading microorganisms. Soil-borne pathogens encounter a first barrier to plant infection as the result of the activity of root border cells [8], which, in certain species, produce an extracellular matrix composed of proteins, polysaccharides, and DNA. Pea root border cells, for example, release DNA-containing extracellular traps that immobilize and kill *R. solanacearum* cells [9]. As a result of plant-pathogen co-evolution, however, *R. solanacearum* has acquired the ability to secrete DNases to degrade DNA traps, enabling bacteria to evade this defensive mechanism [9].

A second impediment to pathogen invasion is posed by the evolved ability of plants to perceive various microbial molecules, a recognition that leads to the activation of immune responses. Conserved pathogen-/microbe-associated molecular patterns (PAMPs/MAMPs) can be detected at the plant cell surface by plasma membrane-localized receptors named pattern-recognition receptors (PRRs) [10]. PRR-triggered immunity (PTI) is sufficient to fend off multiple microbes [11], conferring resistance to most potential pathogens. A conserved 22-amino acid peptide of bacterial flagellin (the main protein building the bacterial flagellum), termed flg22, is the best-studied bacterial PAMP in plants; the PRR FLAGELLIN-SENSING 2 (FLS2) can recognize flg22 and trigger a signaling cascade that ends in the activation of PTI [12,13]. Another well-studied bacterial PAMP is the elongation factor *Tu* (or its elicitor peptide elf18), which is perceived by the EF-Tu receptor (EFR) [14,15].

To date, little is known about the perception of PAMPs from *R. solanacearum*. *R. solanacearum* flg22 (flg22^Rsol^) presents sequence polymorphisms that make it undetectable for all plants tested to date, including Arabidopsis, tomato, pepper, eggplant, tobacco, and *Nicotiana benthamiana* [16,17,18]. Certain plants from the *Solanaceae* family have evolved to detect a flagellin epitope other than flg22, named flgII-28 [19]; nevertheless, they do not perceive the polymorphic flgII-28 from *R. solanacearum* either [18]. These observations suggest that *R. solanacearum* flagellin has efficiently evolved to avoid perception by plant PRRs, while preserving its function in bacterial motility, hence representing an exception to most known cases of plant pathogenic bacteria.

Other *R. solanacearum*-derived molecules, nevertheless, have been shown to effectively elicit immune responses in certain plant species. Several species from the *Solanaceae* family, including tomato, *N. benthamiana*, and tobacco, perceive csp22^Rsol^, a peptide derived from *R. solanacearum* cold-shock protein [18]. Interestingly, roots from tomato and *N. benthamiana* plants display immune responses upon treatment with csp22^Rsol^, indicating that *R. solanacearum* likely confronts perception during root invasion. Additionally, elf18 from *R. solanacearum* is able to elicit immune responses in Arabidopsis [20]. Although the csp22 receptor, CORE, is restricted to certain *Solanaceae* species [21] and the elf18 receptor, EFR, is restricted to *Brassicaceae* plants [15], their transgenic expression in other susceptible hosts enhances resistance to *R. solanacearum* [18,20,22], suggesting that the inter-family transfer of PRRs is a promising strategy to generate crops resistant to bacterial wilt disease. Moreover, a recent report has shown that, although the initial invasion by *R. solanacearum* does not lead to PTI activation, the subsequent damage and cell death associated with root colonization (see below) triggers an upregulation of PTI responses in neighboring cells [23]. Whether the effect of perception mediated by PRRs leads to a halt in bacterial infection at the level of root colonization or, on the contrary, is limited to the post-colonization stages still remains to be dissected.

To counter plant defense responses, *R. solanacearum* can deliver effector proteins inside plant cells via a type-III secretion system (T3SS; [24]). These effector proteins, called type-III effectors (T3Es), constitute major virulence determinants in most Gram-negative bacterial pathogens; T3Es from plant pathogenic bacteria are able to manipulate plant cellular functions and alter plant signal transduction, allowing the bacteria to proliferate, and give rise to the development of disease symptoms [25,26,27]. At the same time, and as a result of the arms race between hosts and pathogens, T3Es can also be detected in some plant species by intracellular receptors containing nucleotide-binding and leucine-rich repeat domains (NLRs), resulting in the activation of immune responses that effectively hinder pathogen proliferation as part of the so-called effector-triggered immunity (ETI) [28]. Several T3Es from *R. solanacearum* have been reported to elicit immune responses in different host plants [29,30,31,32,33,34,35,36,37]. While in some cases T3E recognition leads to an incompatible interaction and the subsequent restriction of host range [31], in other cases these immune responses are further suppressed by other T3Es in the same strain [37], reflecting the bacterial adaptation to T3Es recognition.

Although the results mentioned above indicate that natural resistance to *R. solanacearum* exists, this soil-borne pathogen has an unusually broad host range, indicating that it has efficiently evolved to avoid recognition and/or suppress downstream signaling and responses in multiple plant families. In this regard, it is noteworthy that the T3E repertoire of *R. solanacearum* is particularly large in comparison with other bacterial pathogens, containing more than 70 different T3Es in specific strains [38,39]. Although this inevitably entails a potential risk in the form of T3E recognition, it also confers *R. solanacearum* a significant potential for the suppression of immunity and the manipulation of plant cellular functions in different host plants.

## 3. The Establishment of *R. solanacearum In Vitro* Inoculation Assays Paved the Way to the Study of the Early Stages of the Infection

### 3.1. Classic Soil-Drenching Assay

Given the soil-borne nature of *R. solanacearum*, soil drenching with a bacterial suspension is the most widely used inoculation method to study *R. solanacearum*-plant interactions [40]. It is generally accepted that this method recreates the natural infection process, where the roots will be exposed to a bacterial inoculum from an external source. Upon inoculation, wilting symptoms associated with the development of the disease are scored and represented over time [40]. Several modifications of this method have been developed over the years to add accuracy and allow for quantitation, either by counting bacterial colonies upon serial dilution plating, or by determining the bacterial concentration using luminescence or quantitative PCR (qPCR) of the unique endoglucanase gene (*egl1*) in the *R. solanacearum* chromosome [40,41]. Although soil drenching-based inoculation assays have been extremely useful to help us understand *R. solanacearum*-plant interactions, they entail several limitations: the relatively uncontrolled conditions of the soil micro-environment and the high variability caused by infection randomness may hinder the accurate study of early stages of the infection process; moreover, soil opacity makes it difficult to observe morphological changes in plant roots associated with the bacterial invasion.

### 3.2. In Vitro Inoculation Assays and their Application to Observe Root Development during the Infection by R. solanacearum

The development of experimental systems to perform *R. solanacearum* inoculation in plants grown in vitro facilitated the study of the early stages of the infection, and has been applied to different plant species, including tomato [3], petunia [42], *M. truncatula* [4,6], and Arabidopsis [5,43]. In general, the observations made using this system allowed to divide the bacterial infection of the root in three stages. In the first stage, bacterial cells attach to the root surface and initiate tissue invasion, as shown in Figure 1A. In tomato, bacteria have been found to enter the root through wound sites or natural openings, such as lateral root emerging sites or the axils of secondary roots [3]; however, in *M. truncatula* and Arabidopsis seedlings grown *in vitro*, *R. solanacearum* can penetrate roots between the epidermal cells at the root apex [4,5]. The second stage of the invasion involves the centripetal movement of bacterial cells, aimed at reaching the vascular system, and, in some cases, bacterial multiplication in the intercellular spaces, as displayed in Figure 1B. In tomato and *M. truncatula*, *R. solanacearum* replicates extensively in the periplasm of epidermal and cortical cells, and causes massive plasmolysis [3,4,44]. Nevertheless, the connection between cell lysis and bacterial multiplication is unclear, since in Arabidopsis seedlings *R. solanacearum* causes plasmolysis of epidermal, cortical, and endodermal cells, but it does not seem to significantly replicate in these tissues [5]. In fact, in this experimental system, plasmolysis seems to occur in plant cells that are not in direct contact with bacterial cells, ultimately leading to the collapse of the whole cortex [5]. In the third and last stage of invasion, *R. solanacearum* enters the vascular cylinder and colonizes the xylem, as shown in Figure 1B. In Arabidopsis seedlings, *R. solanacearum* first causes the enlargement of endodermal cells, leading to their breakdown, followed by collapse of the pericycle cells located at the xylem poles [5]. Subsequently, the induced degradation of the cell wall of protoxylem cells allows bacteria to access the xylem, where they proliferate and move through xylem vessels, eventually clogging the xylem space and causing the characteristic wilting symptoms associated with the disease, as displayed in Figure 1C. It is worth noting that, while in vitro inoculation assays are useful to observe root development, they may also produce artifacts in plant and/or bacterial responses, due to the intrinsic characteristics of the experimental system (e.g., high humidity).

## 4. Root Morphological Alterations upon *R. solanacearum* Inoculation

It is likely that, upon contact with the root cells, *R. solanacearum* is recognized by the plant and induces the activation of defence responses. Importantly, it is generally accepted that the activation of defence can have a dramatic effect on plant development, although the exact underlying mechanisms are unclear. Additionally, the observation that the root invasion by *R. solanacearum* entails morphological changes in specific cell types, as outlined above, makes it conceivable that the overall development of the root will be altered by the invasion by this pathogenic bacterium, either as a consequence of the activation of anti-bacterial responses, or following active manipulation by the pathogen.

Indeed, several macroscopic alterations are readily observed upon in vitro inoculation of roots with *R. solanacearum*. Among them, the inhibition of primary root growth is a common phenomenon reported in different plant species [5,6,42,43]; this growth arrest is likely caused, at least partly, by the aforementioned cell death at the root tips and the concomitant disruption of the activity of the root apical meristem [5,6,43]. At least in *M. truncatula* and in petunia, early root alterations include swelling of the root tip [6,42]. Interestingly, although early *R. solanacearum* infection inhibits the formation of lateral roots in Arabidopsis and petunia [5,42], it seems to promote the formation of other root structures. In petunia, *R. solanacearum* induces the formation of emerging bulges at the main root and elongated lateral roots; these lateral root structures (LRSs) derive from the division of pericycle founder cells with abnormal morphology, and constitute efficient colonization sites where bacteria replicate massively [42]. However, and somewhat surprisingly, the formation of LRSs is not required for bacterial invasion of the vascular system or plant colonization [42], suggesting that either they are a mere side effect of, or they may contribute to, other processes in the *R. solanacearum* life cycle. In Arabidopsis seedlings, *R. solanacearum* has been shown to induce the formation of root hairs at the root tip maturation zone [43], and, upon prolonged exposure to the bacteria, the generation of secondary roots [45], both of which may offer additional bacterial penetration sites, hence potentially having a positive effect at the microbial population level. Abnormal cell maturation has been observed in *M. truncatula* roots upon *R. solanacearum* infection, including the formation of mature xylem vessels close to the root tip [6]; the promotion of xylem maturation could benefit the bacterial infection by allowing for earlier, more efficient, and organism-wide colonization. Altogether, a growing body of results demonstrates that *R. solanacearum* infection impacts different aspects of root development, although whether these alterations are the result of direct virulence activities aimed at promoting bacterial pathogenesis or are just a collateral effect of the events enabling or derived from the microbial invasion remains to be determined.

In favour of the former, various virulence factors have been shown to contribute to the different root morphological alterations. The major pathogenicity determinant in *R. solanacearum* is the T3SS and the T3Es injected into plant cells [46]. The bacterial protein HrpB regulates the expression of the components of the T3SS and of the T3Es, while the central virulence regulator HrpG controls the expression of HrpB and other pathogenicity determinants [47]. Interestingly, although both *hrpB* and *hrpG* mutants are unable to induce the formation of root hairs in Arabidopsis, the *hrpB* mutant still inhibits root growth [43]. Similarly, in petunia, an *hrpB* mutant is still able to inhibit root growth and cause swelling of the root tip, but it does not induce the formation of LRSs [42]. These findings indicate that the plethora of root alterations observed during the invasion by *R. solanacearum* do not have a common origin, and only some of them depend on the activity of T3Es. In line with this, in *M. truncatula*, two T3Es, Gala7 and AvrA, are shown to be required for the development of root epidermal cell death, although only Gala7 is necessary for the eventual development of disease symptoms [6]; the specific molecular activities underpinning these effects, nevertheless, remain elusive. Although several other T3Es are known to contribute to different steps of the plant colonization, such as the attachment of bacterial cells to roots, root invasion, and bacterial proliferation inside plant tissues, the specific contribution of other members of the T3E repertoire to the development of certain root morphological alterations is in need of further investigation. On the other hand, the trigger of the T3SS-independent developmental changes is at this point unclear, although defence responses or the action of other virulence factors, such as toxins, could be hypothesized as causative.

## 5. Root Transcriptome Reprogramming during *R. solanacearum* Infection May Contribute to Root Morphological Alterations

Transcriptional reprogramming is a common event in plant-pathogen interactions, resulting from the recognition of the biotic threat by the plant as well as from the manipulation of the plant cell by the invading pathogen. Several recent studies have set out to determine root transcriptional responses during *R. solanacearum* infection in different plant species, including Arabidopsis [45], tomato [48], or the wild potato *Solanum comersonii* [49]. In Arabidopsis, the transcription of several key regulators of root architecture is reported as significantly altered by the presence of the bacteria, suggesting that the morphological alterations caused by *R. solanacearum* infection may have a transcriptional basis [45]. For instance, the expression of genes involved in auxin biosynthesis, transport, and signaling is upregulated upon *R. solanacearum* inoculation in Arabidopsis seedlings [45]. Auxin is a major regulator of root development [50]; strikingly, *R. solanacearum* does not induce the formation of root hairs in the Arabidopsis auxin-insensitive *tir1* mutant, although this mutant still shows inhibition of root growth and cell death at the root tip after bacterial inoculation [43]. Supporting a role of auxin signalling in determining the ability of *R. solanacearum* to manipulate plant development and effectively infect the plant, the transcriptional response of roots of a tomato resistant cultivar displays a relative downregulation of auxin-related genes when compared to a susceptible cultivar [48]. This same study showed that a tomato *dgt1-1* mutant, which has altered auxin transport and impaired formation of secondary roots, exhibits enhanced resistance to *R. solanacearum* [48]. Therefore, it is tempting to hypothesize that auxin-mediated morphological alterations actively contribute to the success of *R. solanacearum* infection; nevertheless, it is worth noting that the *dgt1-1* mutant also displays enhanced resistance upon stem inoculation of the bacteria, which bypasses the root penetration process, therefore raising the possibility that auxin plays a role in later stages of the plant-bacteria interaction [48]. In contrast with the results observed in tomato, auxin-related genes were upregulated in both resistant and susceptible wild potato cultivars upon inoculation with *R. solanacearum* [49]. Notably, reports in these three plant species, namely Arabidopsis, tomato, and wild potato, highlight an alteration of hormone signaling pathways upon contact with the pathogen, which may affect root architecture and have direct or indirect effects on the bacterial invasion. In Arabidopsis roots, infection by *R. solanacearum* also enhances ABA-responsive gene expression, and a multiple ABA receptor mutant impaired in ABA perception is shown to be more susceptible to *R. solanacearum* infection [45]. However, this mutant also displays normal root morphological changes associated with *R. solanacearum* infection [45], which suggests that, despite contributing to resistance against *R. solanacearum*, ABA is not required for the concomitant alterations in root architecture, and affects bacterial virulence either downstream of or in parallel to these changes. In addition to well-known genes in the auxin and ABA signalling and biosynthetic pathways, several other genes with described roles in root development are transcriptionally altered at different stages of the root infection by *R. solanacearum* [45]. These include genes associated with root growth control (e.g., *CLE20*, *PXMT1*, *LRP1*), hair formation and elongation (e.g., *ZFP5*, *OXI1*, *LRX1*), and the emergence of secondary roots (e.g., *CLE1*, *CLE3*, and *GLIP2*), among others. Further detailed studies will be required to determine the degree of involvement of these genes in the alteration of root development and disease caused by *R. solanacearum*.

## 6. Conclusions

*R. solanacearum*, causal agent of bacterial wilt, a devastating crop disease worldwide, is a soil-borne pathogen that infects host plants through the root system. There are multiple fascinating examples in nature of pathogens evolving to effectively manipulate the development of their hosts in order to promote their own replication and spread. Interestingly, a growing body of evidence indicates that *R. solanacearum* triggers developmental abnormalities in the roots of the plants it infects, some of which may benefit bacterial invasion of plant tissues and subsequent spread throughout the host. However, and although it is tempting to speculate that this pathogenic bacterium actively manipulates plant development to promote its own virulence, we are still far from a complete understanding of how these changes come to be, in the first place, and what their effect on bacterial performance, if any, is. In the past few years, detailed analyses at the organismal, tissue, cellular, and molecular levels have shed light on the processes concomitant to the successful colonization of the root by this bacterial species; however, further studies and a careful dissection of the different aspects of the initial stages of the plant-bacteria interaction will be essential to fully elucidate the events and requirements underlying the root invasion by *R. solanacearum*.

## Figures and Tables

**Figure 1 plants-09-00516-f001:**
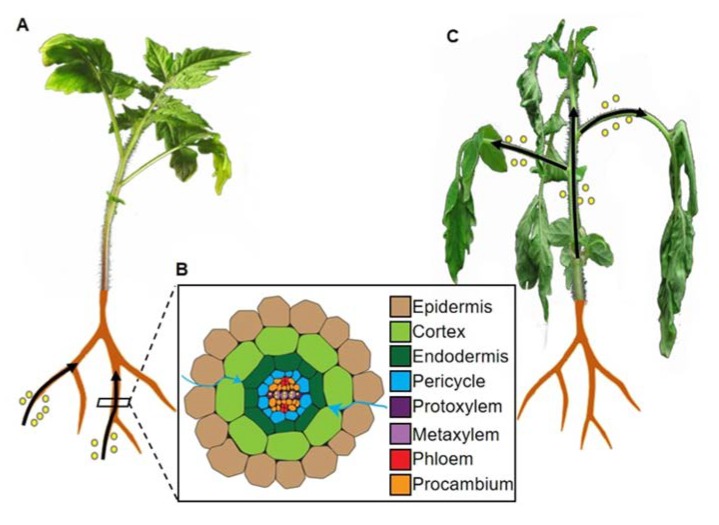
Plant invasion by the pathogenic bacterium *Ralstonia solanacearum.* (**A**). In the first stage of invasion, bacteria (depicted as yellow circles) enter roots through wounds, emerging lateral roots, and root tips. (**B**). Schematic representation of a cross-section of a root; different cell types are indicated. In the second stage of invasion, bacteria massively multiply in the intercellular spaces between cortex cells (blue arrows), and cause plasmolysis of epidermal cells. (**C**). In the last stage of invasion, bacteria (depicted as yellow circles) move throughout the plant through xylem vessels, causing clogging of the vascular system and the typical wilting symptoms. Bacterial movement is depicted as black arrows.

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
