# Peer review of "Insights into the Root Invasion by the Plant Pathogenic Bacterium Ralstonia solanacearum"

_plants, 2020, doi:10.3390/plants9040516_

Round 1
Reviewer 1 Report
The manuscript by Xue et al. is well written and provides a concise overview of the initial steps of Ralstonia solanacearum infections. Despite the wealth of literature on R. solanacearum, the authors manage to keep the focus on specific aspects, such as the role of plant defences and type-3 effector proteins, alteration of root physiology and transcriptional changes in the light of Ralstonia infection.
The authors cite important studies that allowed conclusions about genetic and developmental processes underlying plant-pathogen interaction. I especially enjoyed reading section 5 when it got very specific about auxin-related developmental changes in the root.
While reading the description how Ralstonia invades the root tissue and causes cell death, a recent publication by Zhou et al. 2020, Cell (https://doi.org/10.1016/j.cell.2020.01.013) came to my mind and would be worth including in the review. The authors of this study found that damage enhances local PTI responses in the root upon Ralstonia infection.
Line 134: While the authors highlight the limitations of soil-based assays, they could also mention the possibility that in vitro assays may produce artefacts in plant and/or bacterial responses (e.g. due to high humidity).
Line 225: mention that this was done in tomato, because it was not immediately clear from the text.
Typo errors:
Line 104: and
Line136: in in
Author Response
Reviewer 1:
>>We thank the reviewer for the positive feedback and useful suggestions. Please find below a point-by-point response:
The manuscript by Xue et al. is well written and provides a concise overview of the initial steps of Ralstonia solanacearum infections. Despite the wealth of literature on R. solanacearum, the authors manage to keep the focus on specific aspects, such as the role of plant defences and type-3 effector proteins, alteration of root physiology and transcriptional changes in the light of Ralstonia infection.
The authors cite important studies that allowed conclusions about genetic and developmental processes underlying plant-pathogen interaction. I especially enjoyed reading section 5 when it got very specific about auxin-related developmental changes in the root.
While reading the description how Ralstonia invades the root tissue and causes cell death, a recent publication by Zhou et al. 2020, Cell (https://doi.org/10.1016/j.cell.2020.01.013) came to my mind and would be worth including in the review. The authors of this study found that damage enhances local PTI responses in the root upon Ralstonia infection.
>>We thank the reviewer for this suggestion. We have included discussion associated to this manuscript in the text (Lines 93-96).
Line 134: While the authors highlight the limitations of soil-based assays, they could also mention the possibility that in vitro assays may produce artefacts in plant and/or bacterial responses (e.g. due to high humidity).
>>We thank the reviewer for this suggestion. We have included this consideration in the manuscript (Lines 166-168).
Line 225: mention that this was done in tomato, because it was not immediately clear from the text.
>>The statement in Line 225 actually refers to Arabidopsis. This sentence and the nearby sentences have been revised to specify this (Lines 235-239). The observations in tomato are described below Line 240.
Typo errors:
Line 104: and
Line136: in in
>> We thank the reviewer for pointing these out. These typos have been corrected in the text.

Reviewer 2 Report
The review is relatively well written mainly in terms of changing root morphology and transcription in the root when Ralstonia solanacearum naturally infects the host plants through roots. In this sense, the submitted review article is suitable for publication in plants.
Section 2 summarizes the well-known plant defense response against PAMPs. In this sense, it just follows the previous reviews. Sections 3 to 5 are quite unique in terms of focusing on the root morphology. Section 3.2 describes the alternative inoculation method to observe the early stages of bacterial infection. Since the intercellular spaces in the root are too small for bacteria to multiply much. So the infected bacteria could collapse the cortex to increase the space for multiplication.
Section 4 shows a detail description of root morphological alterations with Ralstonia infection. Although Ralstonia infection impacts different aspects of root development, as the authors pointed out, there are no clear answer to the question if the root alterations could affect the virulence of bacteria.
Section 5 is focusing on plant hormones in the case of bacteria infection. Salicylic acid and jasmonic acid are well known hormones involved in plant defense response to bacteria. In section 5, other hormones, auxin and ABA, are mentioned in details in response to bacteria infection. In this sense, the review is valuable.
One minor thing is in line 127. Please use small and italic letters for the gene name, EGL1.
Author Response
Reviewer 2:
>>We thank the reviewer for the positive feedback and useful suggestions. Please find below a point-by-point response:
The review is relatively well written mainly in terms of changing root morphology and transcription in the root when Ralstonia solanacearum naturally infects the host plants through roots. In this sense, the submitted review article is suitable for publication in plants.
Section 2 summarizes the well-known plant defense response against PAMPs. In this sense, it just follows the previous reviews. Sections 3 to 5 are quite unique in terms of focusing on the root morphology. Section 3.2 describes the alternative inoculation method to observe the early stages of bacterial infection. Since the intercellular spaces in the root are too small for bacteria to multiply much. So the infected bacteria could collapse the cortex to increase the space for multiplication.
Section 4 shows a detail description of root morphological alterations with Ralstonia infection. Although Ralstonia infection impacts different aspects of root development, as the authors pointed out, there are no clear answer to the question if the root alterations could affect the virulence of bacteria.
Section 5 is focusing on plant hormones in the case of bacteria infection. Salicylic acid and jasmonic acid are well known hormones involved in plant defense response to bacteria. In section 5, other hormones, auxin and ABA, are mentioned in details in response to bacteria infection. In this sense, the review is valuable.
One minor thing is in line 127. Please use small and italic letters for the gene name, EGL1.
>> We thank the reviewer for pointing this out. The egl1 name has been corrected in the text.